# Adverse drug reactions associated with the use of biological agents

**Jorge Enrique Machado-Alba**[1]*, **Anyi Liliana Jiménez-Morales**[2], **Yulieth Carolina Moran-Yela**[2], **Ilsa Yadira Parrado-Fajardo**[3], **Luis Fernando Valladales-Restrepo**[1,4]

1 Grupo de Investigación en Farmacoepidemiología y Farmacovigilancia, Universidad Tecnológica de Pereira-Audifarma S.A, Pereira, Risaralda, Colombia, 2 Universidad Tecnológica de Pereira-Audifarma S.A, Pereira, Risaralda, Colombia, 3 Grupo de Investigación en Farmacoepidemiología y Farmacovigilancia, Universidad Tecnológica de Pereira-Audifarma S.A, Bogota DC, Colombia, 4 Grupo de Investigación Biomedicina, Facultad de Medicina, Fundación Universitaria Autónoma de las Américas, Pereira, Colombia

* machado@utp.edu.co

## Abstract

### Introduction

Biological drugs open new possibilities to treat diseases for which drug therapy is limited, but they may be associated with adverse drug reactions (ADRs).

### Objective

To identify the ADRs associated with the use of biological drugs in Colombia.

### Methods

This was a retrospective study of ADR reports from 2014 to 2019, contained in the database of Audifarma SA pharmacovigilance program. The ADRs, groups of associated drugs, and affected organs were classified.

### Results

In total, 5,415 reports of ADRs associated with biological drugs were identified in 78 Colombian cities. A total of 76.1% of the cases corresponded to women. The majority were classified as type A (55.0%) and B (28.9%), and 16.7% were serious cases. The respiratory tract was the most affected organ system (16.8%), followed by the skin and appendages (15.6%). Antineoplastic and immunomodulatory drugs accounted for 70.6% of the reports, and the drugs related to the greatest number of ADRs were adalimumab (12.2%) and etanercept (11.6%).

### Conclusions

The reporting of ADRs has increased in recent years and these reactions are mostly classified as tyoe A or B, categorized as serious in almost one-fifth of the reported cases and associated mainly with immunomodulators and antineoplastic agents. This type of study can support decision makers in ways that benefit patient safety and interaction with health systems.

**Data Availability Statement:** All relevant data are uploaded to protocols.io and accessible via the

following DOI: dx.doi.org/10.17504/protocols.io.
bkcfkstn.

**Funding:** No. Funding sources. The present study
did not receive funding.

**Competing interests:** Declaration of interest. The
authors declare no conflicts of interest.

## Introduction

Biological drugs are derived from expressed proteins, monoclonal antibodies, vectors (viruses
and lipid molecules), antibody fragments and antisense molecules using innovative genetic
engineering methods and recombinant DNA technology, which then converted into drug
complexes during manufacturing [1]. Adverse drug reactions (ADRs) are events that can seri-
ously affect the health of individuals who take drugs for therapeutic, diagnostic or prophylactic
purposes. Very often, hospital care may be required due to the presentation of undesirable
effects, which may also be responsible for significant mortality [2].

The development and use of biological drugs is booming in most countries, since these
drugs open new possibilities for the treatment of diseases for which drug therapy is limited [3,
4]. They constitute a therapeutic innovation, which also represents an unknown world of
adverse reactions and events that affect patient safety. For this reason, it is necessary to analyze
patient records to identify all undesirable events and detect early signs that reduce patient risk
and to make comparisons with safety profile reports available in international reference enti-
ties so that public warnings can be issued [5]. In addition to endangering the health of individ-
uals, ADRs cause treatment abandonment and unexpected costs that affect the finances of
health systems, so their early identification can help prevent and solve these problems [6, 7]. It
is important to clarify that the term "severe" is used to describe the intensity (severity) of an
ADR (for example, mild, moderate or severe), while the term "serious" is related to events that
represent a threat to the patient's life; therefore seriousness (not severity) serves as a guide for
defining regulatory reporting obligations [8]. Hence, pharmacovigilance is the cornerstone in
monitoring drug safety during clinical use [9].

Because information on the safety associated with the use of biological drugs, the incidence
rates of events and their seriousness, the causality association and the data on the true benefit/
risk ratio are insufficient, our objective was to identify the ADRs related to the use of biological
drugs in patients affiliated with the Colombian Health System between 2014 and 2019.

## Materials and methods

A retrospective study was conducted to analyze the systematized databases of reports of ADRs
and suspected ADRs occurring between January 1, 2014, and December 31, 2019, that were
associated with the biological drugs dispensed by the company Audifarma SA. Audifarma is a
drug-dispensing logistics operator that covers more than 8.5 million users of the Colombian
Health System, corresponding to 17.3% of the population affiliated with it, including patients
under the contributory or employer-paid regime and the state-funded regime.

The reports are usually made by the treating physicians, nurses responsible for patient care,
administrative personnel involved in treatment adherence monitoring or patient support pro-
grams and pharmacists in charge of pharmacotherapeutic monitoring of ADR reports. The
information was processed by the group of pharmaceutical chemists from Audifarma who
received the reports of suspected ADRs, checked the data, input them into the system and ana-
lyzed each report. In addition, support is provided by a pharmacoepidemiologist when needed.
Because the data are typed into the database by different professionals at the national level, the
recorded data were checked and verified, and specific compilations were created for annual
periods from 2014 to 2019. All of the cases received are included in the pharmacovigilance pro-
gram and reported to the National Pharmacovigilance Program of the National Institute of
Drug and Food Surveillance of Colombia (Instituto Nacional de Vigilancia de Medicamentos
y Alimentos—INVIMA) within the established deadlines, including the information required
by current legislation.

Only the records of patients with complete information, case follow-up and causality analysis were included. Incomplete records or records considered null were excluded. The grounds for exclusion included the following: 1. report without associated ADR. 2. duplicate report. 3. medication not dispensed by Audifarma. 4. medication error. 5. quality or nonconforming product complaint; and 6. lack of dates.

The general database included the filing date of the ADR report, city, drug (generic name), drug anatomical therapeutic chemical (ATC) classification (letter code and two digits) [10], seriousness (serious, or not serious) [8], type of ADR according to the Rawlins and Thompson classification (A: augmented pharmacological effect, B: bizarre effects not related to pharmacological effect, C: dose-related and time-related, D: time-related, E: withdrawal, F: unexpected failure of therapy) [11], ADR probability classification according to the World Health Organization (WHO: certain, probable, possible, unlikely, conditional, and unassessable) [12], reported event and the traceability of the INVIMA report submission.

The reported ADRs were standardized according to the WHO adverse reaction terminology (WHO-ART) [13]. The main drugs for which ADRs were reported were classified, describing the first 15 ATC subgroups (letter code and first two digits), and an ADR list was created for the 10 drugs with the highest numbers of reports.

The statistical package SPSS 26.0 for Windows (IBM, USA) was used for data analysis, and the data are expressed as frequencies, percentages and means. Incidence rates were estimated from the ADR reports and total patients who were dispensed biological drugs per monitoring year and per 100,000 health system affiliates.

The present study was approved by the Bioethics Committee of the Universidad Tecnológica de Pereira under the risk-free research category (approval number 0104–2019). The principles established by the Declaration of Helsinki were respected. No personal data of the patients were used.

## Results

A total of 5,415 ADR reports associated with the use of 71 biological drugs were identified throughout the six years of monitoring, in 78 Colombian cities, and with respect to 10 health insurance companies and 65 healthcare institutions, including clinics and hospitals; a progressive increase in the number of cases was observed (Table 1). A total of 4,122 (76.1%) reports corresponded to female patients.

According to the ADR seriousness, 77.4% cases were classified in the nonserious category, followed by serious events, and six were associated with a fatal outcome. In addition, a low percentage (0.2%) could not be classified (Table 1). The drugs associated with lethal ADRs were abatacept (four cases), etanercept (one case) and rituximab (one case). The most common ADR type was type A, followed by type B and type C reactions (Table 1).

According to the ATC classification analysis, antineoplastics and immunomodulators were the groups with the highest number of reports, followed by medications for the respiratory and skeletal muscle systems (Table 2). The therapeutic subgroups most frequently associated with ADRs were immunosuppressants, other antineoplastic agents (including monoclonal antibodies) and drugs for systemic use for obstructive airway diseases (Table 2).

The most common ADRs were those causing respiratory system disorders, followed by skin and appendages disorders and general disorders (Table 3). Causality analysis indicated that most ADRs were considered possibly associated with the reported drug (67.9% were certain, probable or possible) (Fig 1). The biological drugs with the highest number of reports in the monitoring period were adalimumab, etanercept and omalizumab. The drugs whose incidence increased the most between 2014 and 2019 were denosumab (30.0% increase), omalizumab

**Table 1. Number of reports per year, classification according to seriousness and type of adverse reactions in patients treated with biological agents in Colombia from 2014–2019.**

|  | Number of cases (n = 5,500) | Percentage |
|---|---|---|
| **Year of report** |  |  |
| 2014 | 187 | 3.5 |
| 2015 | 211 | 3.9 |
| 2016 | 360 | 6.6 |
| 2017 | 864 | 16.0 |
| 2018 | 1,321 | 24.4 |
| 2019 | 2,472 | 45.7 |
| **Classification according to seriousness** [8] |  |  |
| Not serious | 4,192 | 77.4 |
| Serious | 903 | 16.7 |
| Lethal | 6 | 0.1 |
| Therapeutic failure | 317 | 5.9 |
| Not classified | 9 | 0.2 |
| **Type of adverse reaction** [11] |  |  |
| A (Augmented) | 2,925 | 55.0 |
| B (Bizarre) | 1,563 | 28.9 |
| C (Chronic) | 152 | 2.8 |
| D (Delayed) | 87 | 1.6 |
| E (End of treatment) | 4 | 0.1 |
| F (Failure) | 209 | 3.9 |
| Without classification | 1,145 | 21.1 |

(18.4%) and etanercept (15.2%). There was an estimated 41.7% increase in the incidence of ADR reports for secukinumab between 2016 and 2019. There were smaller increases for abatacept (3.2%) and rituximab (1.9%). The total number of reports for each of the 10 biological drugs with the highest numbers of ADRs, the percentages they represented among all notifications, the incidences per 100 patients who received them and the incidences compared by 100,000 affiliates between 2014 and 2019 are shown in Table 4.

## Discussion

It was possible to determine which drugs of biological origin were most frequently involved in ADRs in the Colombian population; this aim was the objective of this study. Biological drugs must have specific pharmacovigilance considerations, including closer monitoring that can ensure their effectiveness and safety [14]. Although biological drugs are less commonly used than synthetic drugs (approximately 20% of current drugs are biological drugs), they are very often associated with adverse events, some of which are serious and even lethal [15, 16]. In recent years, reports of biological drugs associated with ADRs have increased worldwide, as also observed in this study, which is perhaps related to greater notification by prescribing physicians, nurses and patients and the increased use of these drugs for the treatment of a large number of pathological entities [4, 17].

ADRs associated with biological drugs occur more frequently in women, as documented in other studies conducted in Spain (82.9%) [18], the United States (75.5%) [19] and Italy (54.3–71.3%) [4, 20], in agreement with the present finding. This phenomenon is probably because many of the pathologies for which biological drugs are used have a known predominance in women, including autoimmune diseases such as rheumatoid arthritis [21, 22] and oncological

**Table 2. Classification of biological agents associated with adverse drug reactions according to the Anatomical Therapeutic Chemical (ATC) group and subgroup in Colombia from 2014–2019.**

| ATC | Description according ATC group | Patients | Percentage |
|---|---|---|---|
| L | Antineoplastic and immunomodulating agents | 3,823 | 70.6 |
| R | Respiratory system | 662 | 12.2 |
| M | Musculo-skeletal system | 351 | 6.5 |
| A | Alimentary tract and metabolism | 265 | 4.9 |
| S | Sensory organs | 137 | 2.5 |
| J | Antiinfectives for systemic use | 88 | 1.6 |
| H | Systemic hormonal preparations, excluding sex hormones and insulins | 52 | 1.0 |
| B | Blood and blood forming organs | 27 | 0.5 |
| V | Various | 4 | 0.1 |
| C | Cardiovascular system | 3 | 0.1 |
| D | Dermatologicals | 3 | 0.1 |
| | **Description according ATC subgroup** | | |
| L04A | Immunosuppressants | 3,201 | 59.1 |
| L01X | Other antineoplastic agents | 562 | 10.4 |
| R03D | Other systemic drugs for obstructive airway diseases | 637 | 11.8 |
| M05B | Drugs affecting bone structure and mineralization | 351 | 6.5 |
| H05A | Parathyroid hormones and analogs | 1 | 0 |
| A16A | Other alimentary tract and metabolism products | 204 | 3.8 |
| A10A | Insulins and analogs | 61 | 1.1 |
| J06B | Immunoglobulins | 88 | 1.6 |
| L03A | Immunostimulants | 60 | 1.1 |
| S01L | Ocular vascular disorder agents | 136 | 2.5 |
| H01A | Anterior pituitary lobe hormones and analogs | 51 | 0.9 |
| B02B | Vitamin K and other hemostatics | 24 | 0.4 |
| R05C | Expectorants, excluding combinations with cough suppressants | 25 | 0.5 |
| S01E | Antiglaucoma preparations and miotics | 1 | 0 |
| C10A | Lipid modifying agents | 3 | 0.1 |

ATC: anatomical therapeutic chemical

diseases [23], which expose women to greater probabilities of biological drug use and of developing ADRs. In the present study, antineoplastics and immunomodulators were the biological drugs most frequently associated with this type of event, in agreement with the findings of Cutroneo et al. in Italy [4].

The ADRs most documented in different studies are those related to infections [19, 20, 24–26], general manifestations the administration of biological drugs [17, 27, 28] and the skin or subcutaneous tissues [4, 17, 28, 29]. However, the present study found that the most common ADRs were those related to the respiratory tract, diverging from what was found in other studies, in which their frequency was much lower (16.8 vs. 3.8–10.8%) [4, 17, 29], probably because we included infections such as pneumonia and bronchitis in this category, among others, which increased the proportion of respiratory tract-related ADRs.

Type A and B ADRs were the most frequent in a previous study conducted in Colombia [30]. That study analyzed a cohort of patients with rheumatoid arthritis treated with synthetic disease-modifying antirheumatic drugs (sDMARD) and biologic disease-modifying antirheumatic drugs (bDMARD) and found that of all ADRs, 87.7% were type A (sDMARD: 70.2%, bDMARD: 17.5%) and 12.3% were type B (sDMARD: 8.1%, bDMARD: 4.2%); and no other

**Table 3. Main systems affected by adverse drug reactions to biological agents in Colombia from 2014–2019.**

| Affected organs | Patients | Percentage |
|---|---|---|
| Respiratory system disorder | 911 | 16.8% |
| Skin and appendages disorder | 845 | 15.6% |
| Body disorder—general | 559 | 10.3% |
| Gastrointestinal system disorder | 379 | 7.0% |
| Musculoskeletal system disorder | 297 | 5.5% |
| Central and peripheral nervous system disorder | 291 | 5.4% |
| General cardiovascular disorder | 194 | 3.6% |
| Urinary system disorder | 189 | 3.5% |
| Vision disorder | 77 | 1.4% |
| Liver and biliary disorder | 71 | 1.3% |
| White blood cell and endothelial reticulum disorder | 43 | 0.8% |
| Application site disorder | 40 | 0.7% |
| Sense organ disorder | 34 | 0.6% |
| Red blood cell disorder | 33 | 0.6% |
| Resistance mechanism disorder | 32 | 0.6% |
| Others | 1,420 | 26.2% |
| | **5,415** | **100.0%** |

ADR types were observed [30]. In addition, according to seriousness, 22.6% of the reports were classified as serious, consistent with what was found in Italy (9.8–25.5%) [20, 28], Japan (18.5–23.4%) [25, 31], Spain (21.7%) [24], Brazil (25.0%) [27] and Korea (32.3%) [17]. Among severe reactions, the possibility of developing cancer, infections, hypersensitivity reactions and major cardiovascular events is described in the literature [15, 17, 19, 24, 27], and fatalities can also occur, which in this report corresponded to 0.1% of all ADRs, a rate lower than that documented in another study [32]. Death does not correspond to an adverse event but rather to a fatal outcome that can also be explained by the underlying disease of the patient. The classification of the severity of the event is independent of the degree of association with its causality.

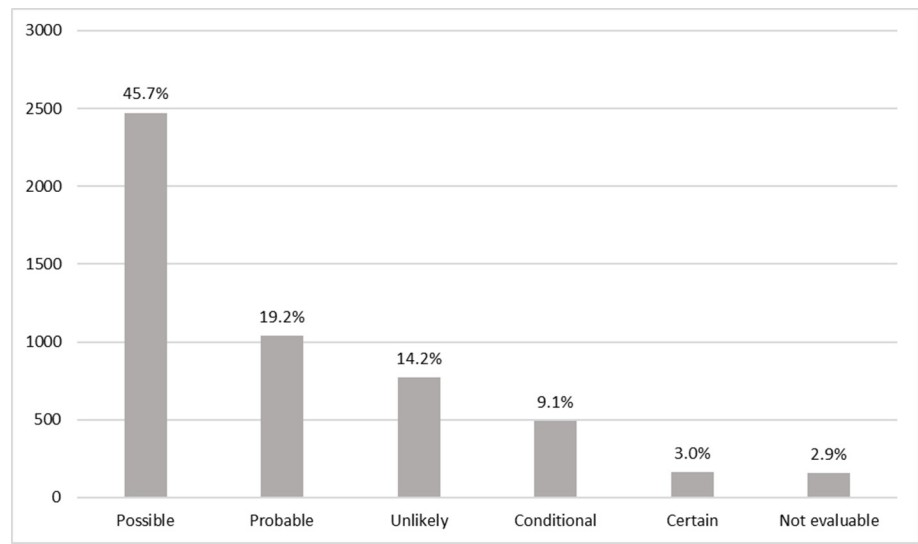

**Fig 1. Adverse drug reactions by biotech agents according to probability classification to the World Health Organization in patients of Colombia.**

**Table 4. Top 10 biological agents with the highest number of reports and their incidences in Colombia from 2014–2019.**

| Biological drug | Number of reported cases | Percentage of all ADR | Mean incidence per 100 patients/ year | Incidence per 100,000 affiliates / year in 2014 | Incidence per 100,000 affiliates / year in 2019 |
|---|---|---|---|---|---|
| Adalimumab | 674 | 12.2 | 6.8 | 0.352 | 3.564 |
| Etanercept | 638 | 11.6 | 7.8 | 0.264 | 4.007 |
| Omalizumab | 612 | 11.1 | 12.5 | 0.176 | 3.241 |
| Tocilizumab | 529 | 9.6 | 10.5 | 0.440 | 3.756 |
| Rituximab | 482 | 8.7 | 7.4 | 0.859 | 1.639 |
| Denosumab | 350 | 6.3 | 1.2 | 0.088 | 2.643 |
| Abatacept | 319 | 5.8 | 7.1 | 0.220 | 0.718 |
| Golimumab | 289 | 5.2 | 12.4 | 0.110 | 0.945 |
| Secukinumab | 176 | 3.2 | 9.4 | 0.031* | 1.292 |
| Ustekinumab | 167 | 3.0 | 6.3 | 0.088 | 0.754 |

* Year 2016. ADR: adverse drug reactions.

In the present study, the main biological drugs related to ADRs were adalimumab and etanercept, in agreement with other studies [18, 20, 27], but the incidence per 100 patients per year was higher than that reported in Spain in patients with rheumatoid arthritis (8.1 for adalimumab and 5.1 for etanercept) [24]. The incidence in our study was determined in a general manner for all ADRs, while the Spanish study considered only serious ADRs [24]. In Brazil, in patients with rheumatoid arthritis and psoriatic arthritis, 55.2% and 19.8% of ADRs were secondary to adalimumab and etanercept, respectively [27]. In Spain, in patients with rheumatoid arthritis, 35.1% and 21.6% of ADRs were due to adalimumab and etanercept, respectively [18]. In Italy, Barbieri et al. studied patients with inflammatory arthritis and found that 27.3% and 19.0% of ADRs were due to etanercept and adalimumab, respectively [20]. However, many studies also report a high proportion of ADRs secondary to infliximab [3, 24, 32], which was not observed in the present study due to the low use of this drug in Colombia. Additionally, the proportion of ADRs secondary to omalizumab in the present study is noteworthy. In Kuwait, in patients with asthma treated with omalizumab, 34.3% had ADRs, and 42.8% discontinued treatment [33].

One of the limitations of this study is its observational nature, as it is based on a database of reports that does not include variables such as patient age, comorbidities and comedications. Additionally, the proportion of patients who had to discontinue treatment due to an ADR was not analyzed, nor were the time elapsed from the administration of the biolgical drug to the onset of the ADR, concomitant medication use or ADRs associated with previous treatments, although all of these factors are identified in the individual report and monitoring of each case. Moreover, for this analysis, no distinction was made between innovative and biosimilar drugs. However, the strongest point of this study is that it compiled ADR reports from one of the largest cohorts of patients in Colombia, for which exhaustive follow-ups were performed to identify the causality association.

## Conclusions

Based on our findings, we conclude that the reporting of ADRs has increased in recent years and that the reactions are mostly classified as type A or B, categorized as serious in almost one-fifth of the reported cases and associated mainly with immunomodulators and antineoplastic agents. It is important to empower physicians and entire health teams to improve the traceability of adverse reactions and thus optimize and strengthen pharmacovigilance programs. This

type of study can support decision makers in aspects that benefit patient safety and interaction with health systems.

## Acknowledgments

We thank Soffy Claritza López, for her work in obtaining the database.

## Author Contributions

**Conceptualization:** Jorge Enrique Machado-Alba, Anyi Liliana Jiménez-Morales, Yulieth Carolina Moran-Yela.

**Data curation:** Anyi Liliana Jiménez-Morales, Yulieth Carolina Moran-Yela, Ilsa Yadira Parrado-Fajardo.

**Formal analysis:** Jorge Enrique Machado-Alba, Anyi Liliana Jiménez-Morales, Yulieth Carolina Moran-Yela.

**Investigation:** Jorge Enrique Machado-Alba, Anyi Liliana Jiménez-Morales, Yulieth Carolina Moran-Yela, Luis Fernando Valladales-Restrepo.

**Methodology:** Jorge Enrique Machado-Alba.

**Project administration:** Jorge Enrique Machado-Alba.

**Supervision:** Jorge Enrique Machado-Alba, Ilsa Yadira Parrado-Fajardo.

**Validation:** Jorge Enrique Machado-Alba, Ilsa Yadira Parrado-Fajardo, Luis Fernando Valladales-Restrepo.

**Writing – original draft:** Jorge Enrique Machado-Alba, Luis Fernando Valladales-Restrepo.

**Writing – review & editing:** Jorge Enrique Machado-Alba.

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
