## [Decision Letter · Decision Letter 0]

17 Nov 2020

PONE-D-20-29482

Adverse drug reactions associated with the use of biological agents

PLOS ONE

Dear Dr. Jorge Enrique Machado-Alba,

Thank you for submitting your manuscript to PLOS ONE. After careful consideration, we feel that it has merit but does not fully meet PLOS ONE’s publication criteria as it currently stands. Therefore, we invite you to submit a revised version of the manuscript that addresses the points raised during the review process.

ACADEMIC EDITOR: 

Please, address the follow reviewers' comments.

We look forward to receiving your revised manuscript.

Kind regards,

Beatrice Nardone

Academic Editor

PLOS ONE

Journal Requirements:

Additional Editor Comments (if provided):

Please, address the follow reviewers' comments.

Reviewers' comments:

Reviewer's Responses to Questions

**Comments to the Author**

1. Is the manuscript technically sound, and do the data support the conclusions?

Reviewer #1: Yes

Reviewer #2: Yes

Reviewer #3: Yes

2. Has the statistical analysis been performed appropriately and rigorously? 

Reviewer #1: Yes

Reviewer #2: Yes

Reviewer #3: Yes

3. Have the authors made all data underlying the findings in their manuscript fully available?

Reviewer #1: Yes

Reviewer #2: Yes

Reviewer #3: Yes

4. Is the manuscript presented in an intelligible fashion and written in standard English?

Reviewer #1: No

Reviewer #2: Yes

Reviewer #3: No

5. Review Comments to the Author

Reviewer #1: Outstanding report and novel and provides very important information regarding the use of biologics and adverse events. This study also provides specific information in Hispanic population that is very important due to the specific reports related with this race. Very much recommended to be publish in the journal for the lack of conflicts on interest that are happening in the last 20 years with big pharmaceutical companies.

Reviewer #2: The article is interesting and brings important information about adverse reactions to biologicals, drugs that are increasingly used. I think it can be accepted for publication after the corrections.

In abstract:

The conclusions item does not answer the objective of the study.

Materials and methods

Line 107-111:

Add the references of the Rawlins and Thompson classification and of ADR probability classification according to the WHO. Put in the table 1 as well.

It would be convenient to explain the Rawlins and Thompson classification (A, B, C, D, E, F) why not all readers know.

Line 112:

"The reported ADRs were standardized according to the WHO adverse reaction terminology (WHO-ART). "Add the references .

The references are not fully in accordance with the journal's rules.

Reviewer #3: Authors needs to revisit the severity assessment in the results sections especially table 1 and also the causality assessment and amend appropriately in the materials and methods section and also in the results section.

6. PLOS authors have the option to publish the peer review history of their article (what does this mean?). If published, this will include your full peer review and any attached files.

Reviewer #1: No

Reviewer #2: No

Reviewer #3: No

---

## [Author Response · Author response to Decision Letter 0]

23 Nov 2020

Pereira, November 23 2020

Dear

Beatrice Nardone

Academic Editor

PLOS ONE

Reference: Manuscript: PONE-D-20-29482 Adverse drug reactions associated with the use of biological agents.

Dear editors,

Thank you very much for your comments. Your observations will certainly improve the quality of the manuscript. We have responded point by point to each of them.

Additional Editor Comments (if provided):

Please, address the follow reviewers' comments.

Reviewers' comments:

Reviewer's Responses to Questions

Comments to the Author

1. Is the manuscript technically sound, and do the data support the conclusions?

Reviewer #1: Yes

Reviewer #2: Yes

Reviewer #3: Yes

A/ Thank you.

2. Has the statistical analysis been performed appropriately and rigorously?

Reviewer #1: Yes

Reviewer #2: Yes

Reviewer #3: Yes

A/ Thank you.

3. Have the authors made all data underlying the findings in their manuscript fully available?

Reviewer #1: Yes

Reviewer #2: Yes

Reviewer #3: Yes

A/ Thank you.

4. Is the manuscript presented in an intelligible fashion and written in standard English?

Reviewer #1: No

Reviewer #2: Yes

Reviewer #3: No

A/ The manuscript was translated into English by experienced translators from American Journal Experts. We submitted these corrections and the new manuscript for a new style review. We have attached the certificate.

5. Review Comments to the Author

Reviewer #1: Outstanding report and novel and provides very important information regarding the use of biologics and adverse events. This study also provides specific information in Hispanic population that is very important due to the specific reports related with this race. Very much recommended to be publish in the journal for the lack of conflicts on interest that are happening in the last 20 years with big pharmaceutical companies.

A/ Thank you.

Reviewer #2: The article is interesting and brings important information about adverse reactions to biologicals, drugs that are increasingly used. I think it can be accepted for publication after the corrections.

A/ Thank you.

In abstract:

The conclusions item does not answer the objective of the study.

A/ We changed the wording of the conclusions to correspond to the stated objective.

Materials and methods

Line 107-111:

Add the references of the Rawlins and Thompson classification and of ADR probability classification according to the WHO. Put in the table 1 as well.

A/ We inserted the relevant references.

It would be convenient to explain the Rawlins and Thompson classification (A, B, C, D, E, F) why not all readers know.

A/ We provided the necessary explanations in the text.

Line 112:

"The reported ADRs were standardized according to the WHO adverse reaction terminology (WHO-ART). "Add the references .

A/ We inserted the relevant references.

The references are not fully in accordance with the journal's rules.

A/ We reviewed each reference to comply with the journal standards.

Reviewer #3: Authors needs to revisit the severity assessment in the results sections especially table 1 and also the causality assessment and amend appropriately in the materials and methods section and also in the results section.

A/ The terms severity and causality were modified in the Methods and Results.

6. PLOS authors have the option to publish the peer review history of their article (what does this mean?). If published, this will include your full peer review and any attached files.

Do you want your identity to be public for this peer review? For information about this choice, including consent withdrawal, please see our Privacy Policy.

Reviewer #1: No

Reviewer #2: No

Reviewer #3: No

A/ Okay.

Reviewer #1: This is a well-conceived study that provides an important assessment of Adverse drug reactions associated with the use of biological agents. This may provide useful information for signal management and benefit-risk assessment.

ABSTRACT

1. Page 2 line 37: Introduction. Biotech drugs open new possibilities to treat diseases for which drug therapy is limited, but they may be associated with serious adverse drug reactions

Change serious adverse drug reactions to adverse drug reactions in line with your title.

A/ The word serious has been deleted.

2. Page 2 line 42-43: The respiratory tract was the most commonly affected organ system (16.8%), followed by the skin and adnexa (15.6%).

Change adnexa (Latin) to appendages (English) as stipulated by MedDRA.

A/ We changed "adnexa" to "appendages".

INTRODUCTION

1. Page 3 line 65-68: Biotech drugs are synthesized from expressed proteins, monoclonal antibodies, vectors (viruses, lipid 66 molecules), antibody fragments, antisense molecules and lipid vectors using innovative genetic engineering methods and recombinant DNA technology, which converts them into drug complexes during manufacturing and these drugs offer the potential to participate in important biological processes in humans [1].

Sentence is too long. Rewrite sentence.

A/ The sentence has been rewritten.

2. Page 3 line 68-70: Side effects and adverse drug reactions (ADRs) are events that can seriously affect the health of individuals who take drugs for therapeutic, diagnostic or prophylactic purposes.

I suggest you stick to adverse drug reaction and not side effect based on your topic and objective

A/ We deleted “side effects”.

3. Page 3 line 82: Because information on the safety associated with the use of biotech drugs, the incidence rates of events and their severity, the causality association and the data on the true benefit/risk ratio are often insufficient, our objective was to identify the ADRs related to the use of biotech drugs in patients affiliated with the Colombian 84 Health System between 2014 and 2019.

Please note that there is a difference between severity and seriousness according to ICH E2A-Clinical Safety data management: Definitions and standards for expedited reporting. Provide clarification.

A/ In the Introduction, clarification regarding the terms severity and seriousness was provided, and we decided to change the word “severity” to “seriousness”.

MATERIALS AND METHODS

1. Page 4 line 92: The reports are usually made by the treating physicians, nurses responsible for patient care, administrative personnel involved in treatment adherence monitoring or patient support programs and pharmaceutical chemists in charge of pharmacotherapeutic monitoring of ADR reports.

I believe you are referring here to Pharmacists. Please note that there is a difference between a pharmacist and a pharmaceutical chemist. So, use the appropriate term if you are referring to pharmacists.

A/ We consider “pharmacists” the correct term.

2. Page 4 line 107-114: The general database included the filing date of the ADR report, city, drug (generic name), drug anatomical therapeutic chemical (ATC) classification (letter code and two digits), severity (serious, not serious), type of ADR according to the Rawlins and Thompson classification (A, B, C, D, E, F), ADR probability classification according to the WHO (definitive, probable, possible, unlikely, conditional, unassessable), reported event and the traceability of report submission to INVIMA. The reported ADRs were standardized according to the WHO adverse reaction terminology (WHO-ART). The main drugs for which ADRs were reported were classified, describing the first 15 ATC subgroups (letter code and first two digits), and an ADR list was created for the 10 drugs with the highest numbers of reports.

a. Provide references and versions for the classifications based on:

i. ATC

ii. WHO-ART

iii. Severity

iv. Causality

v. Types of ADR

A/ Each of the classifications has been cited.

b. Please note that there is a difference between severity and seriousness according to ICH E2A- Clinical Safety data management: Definitions and standards for expedited reporting. Seriousness is into non-serious and serious and the reasons for seriousness ranges from death, congenital anomaly, disability, hospitalization etc. Severity on the other hand based on the type of scale can be into mild, moderate or severe or grade 1, grade 2 grade 3 etc.

A/ In the Introduction, clarification regarding the terms severity and seriousness was provided, and we decided to change the word “severity” to “seriousness”.

c. WHO-UMC system for causality is into certain, probable, possible, unlikely, conditional and unaccessible. Take note and review appropriately

A/ The correction was made.

RESULTS

1. Page 5: Table 1. Number of reports per year, classification according to severity and type of adverse reactions in 127 patients treated with biological agents in Colombia 2014-2019

a. Your classification based on severity as stated in the table is flawed as I stated previously according to ICH E2A. Clarify whether you are classifying based on severity or seriousness.

A/ We changed “severity” to “seriousness”.

b. Under types of adverse reactions, review whether F stands for Failure or Foreign.

A/ We changed “Foreign” to “Failure”.

2. Page 7 line 145-146: The causality analysis determined that most ADRs were considered possibly associated with the reported drug (67.9% were definitive, probable or possible) (Figure 1).

a. Rewrite this sentence and replace definitive with certain according to the WHO-UMC causality assessment system

A/ The correction was made; “definitive” was replaced by “certain”.

b. Review Figure 1 since the WHO-UMC causality assessment analysis and terms does not take into consideration therapeutic failure and not evaluable

A/ We deleted “therapeutic failure” and changed “definitive” to “certain”.

GENERAL COMMENT:

This manuscript could benefit from a professional manuscript editing to improve its grammatical, spelling and intellectual clarity.

A/ This manuscript was translated into English by experienced translators from American Journal Experts. We submitted these corrections and the new manuscript for a new style review. We have attached the certificate.

The authors

---

## [Editor Report · Decision Letter 1]

25 Nov 2020

Adverse drug reactions associated with the use of biological agents

PONE-D-20-29482R1

Dear Dr. Machado-Alba,

We’re pleased to inform you that your manuscript has been judged scientifically suitable for publication and will be formally accepted for publication once it meets all outstanding technical requirements.

Kind regards,

Beatrice Nardone

Academic Editor

PLOS ONE

---

## [Editor Report · Acceptance letter]

7 Dec 2020

PONE-D-20-29482R1 

Adverse drug reactions associated with the use of biological agents 

Dear Dr. Machado-Alba:

I'm pleased to inform you that your manuscript has been deemed suitable for publication in PLOS ONE. Congratulations! Your manuscript is now with our production department. 

Kind regards, 

on behalf of

Dr. Beatrice Nardone 

Academic Editor

PLOS ONE